# Brucellosis and One Health: Inherited and Future Challenges

**DOI:** 10.3390/microorganisms11082070

**Published:** 2023-08-11

**Authors:** Ignacio Moriyón, José María Blasco, Jean Jacques Letesson, Fabrizio De Massis, Edgardo Moreno

**Affiliations:** 1Microbiology and Parasitology Department, Medical School, Universidad de Navarra, 31008 Pamplona, Spain; 2Departamento de Ciencia Animal, Centro de Investigación y Tecnología Agroalimentaria de Aragón (CITA), 50059 Zaragoza, Spain; jblasco@unizar.es; 3Research Unit in Biology of Microorganisms, Narilis, University of Namur, 5000 Namur, Belgium; jean-jacques.letesson@unamur.be; 4Istituto Zooprofilattico Sperimentale dell’Abruzzo e del Molise, 64100 Teramo, Italy; f.demassis@izs.it; 5Programa de Investigación en Enfermedades Tropicales, Escuela de Medicina Veterinaria, Universidad Nacional, Heredia 40104, Costa Rica; emoreno@una.cr

**Keywords:** One Health, brucellosis, awareness, capacity building, climate, global warming, intensification, diagnosis, vaccines

## Abstract

One Health is the collaborative efforts of multiple disciplines to attain optimal health for people, animals and the environment, a concept that historically owes much to the study of brucellosis, including recent political and ethical considerations. Brucellosis One Health actors include Public Health and Veterinary Services, microbiologists, medical and veterinary practitioners and breeders. Brucellosis awareness, and the correct use of diagnostic, epidemiological and prophylactic tools is essential. In brucellosis, One Health implementation faces inherited and new challenges, some aggravated by global warming and the intensification of breeding to meet growing food demands. In endemic scenarios, disease awareness, stakeholder sensitization/engagement and the need to build breeder trust are unresolved issues, all made difficult by the protean characteristics of this zoonosis. Extended infrastructural weaknesses, often accentuated by geography and climate, are critically important. Capacity-building faces misconceptions derived from an uncritical adoption of control/eradication strategies applied in countries with suitable means, and requires additional reference laboratories in endemic areas. Challenges for One Health implementation include the lack of research in species other than cattle and small ruminants, the need for a safer small ruminant vaccine, the need to fill in the infrastructure gap, the need for realistic capacity-building, the creation of reference laboratories in critical areas, and the stepwise implementation of measures not directly transposed from the so-called developed countries.

## 1. Introduction

Brucellosis is the name given to a group of highly contagious zoonotic infections caused by members of the genus *Brucella*. Historically linked to ruminants, swine, and dogs, some brucellae infect camelids, seal and cetacean species, some amphibians, foxes, desert rodent species, cave-dwelling bats and other wild animals [1]. At least in domestic livestock, brucellosis is a significant cause of abortion and fertility loss, severely curtailing animal production. Humans contract the disease from infected animals and their products, and human-to-human transmission is only anecdotic. Thus, risk groups are breeders and their families, veterinarians, laboratory personnel and dairy and slaughterhouse workers. The general public is mainly affected by consuming raw milk and unpasteurized dairy products and, to a lesser extent, raw viscera, blood and offal [2]. Human brucellosis is a severe and debilitating disease that may leave permanent sequelae if untreated, requiring prolonged, expensive and combined antibiotic therapy. However, for cases caused by contact with infected wild-life, brucellosis is a traveler-disease in the handful of countries that eradicated this disease from domestic livestock and strictly implement milk pasteurization. On the other hand, although the actual incidence is unknown [3,4], brucellosis is considered a severe problem in animals and humans in Asia, the Middle East, Africa and Ibero-America, as illustrated in prioritization analyses [5]. Indeed, this is consistent with the presence of *Brucella* species in many countries registered in the World Animal Health Information System (WAHIS) [6].

Because of the wide range of hosts, various routes of transmission, the impact on animal and human welfare, and the conditions and worldwide distribution of the countries affected, it is not surprising that One Health in brucellosis has been the topic of at least 10 publications, either in general or when applied to particular situations [7,8]. With variable emphasis, these works summarize the parasite’s biology, transmission patterns, control and vaccination measures and levels of intersectoral cooperation. Here, we will complement these previous works by considering first to what extent the One Health paradigm changes our perspective of this zoonosis. Then, insofar as is possible, we will document the challenges faced by implementing One Health, those common to other infectious diseases and those specific to brucellosis that are not sufficiently discussed in previous works. We exclude general aspects of *Brucella’s* biology, epidemiology and pathogenesis that only indirectly pertain to One Health and refer the reader to well-documented reviews [9,10].

## 2. Is One Health a New Concept in Brucellosis?

Whereas One Health is commonly presented as a new perspective in the control and eradication of brucellosis, to a great extent, it was the other way around: the One Health paradigm owes much to the study of this zoonosis in the century [11]. Nearly 30 years ago, Calvin Schwabe developed the concept of “One Medicine”, later expanded and renamed One Health, as necessary to face the “immensely complex, multifaceted future quality of human life and, ultimately, of human survival” (quoted in [12]). Notably, he used brucellosis to illustrate the validity of the “One Medicine” perspective. Similarly, James Steele, the long-recognized father of Veterinary Public Health, was among the first scientists to advocate what we now name One Health principles in zoonotic disease management [13], undoubtedly influenced by his familiarity with brucellosis. Significantly, the rapid appreciation of Schwabe’s “One Medicine” also included Wesley Spink [13], a key figure in the field of brucellosis whose 1956 book “The Nature of Brucellosis” remains an invaluable source of information about many aspects of the disease, including, in Chapter 1, the historical perspective on which One Health brucellosis is grounded [14]. To these inputs and advocacy, we have to add the experience of some European countries, the USA, Canada, and Australia, which progressively adopted One Health control and eradication policies for brucellosis before this name was coined. Indeed, the validity of these experiences and the need for this conceptual approach are confirmed by the success and difficulties encountered in the control of brucellosis and other zoonoses in Malta, Serbia, Greece, Kenya, Ethiopia, Uganda, Israel, Jordan, Mongolia, Azerbaijan and India [15,16,17,18,19,20,21,22].

An abridged definition of One Health corresponds to the collaborative efforts of multiple disciplines to attain optimal health for people, animals and the environment [13]. Of course, more detailed descriptions of the One Health concept exist, many by specific experts and international agencies and bodies, including the Food and Agriculture Organization of the United Nations (FAO), The World Organization for Animal Health (WOAH, formerly OIE), the World Health Organization (WHO), and the World Bank (reviewed in [17]). It is also worth mentioning that this concept overlaps with other global concerns: recently, the United Nations Environment Programme (UNEP) signed an agreement with WHO, FAO and WOAH to strengthen cooperation to sustainably balance and optimize the health of humans, animals, plants and the environment. The elements used in these definitions and their interconnections are summarized in Figure 1 [23], which illustrates the complexities of a holistic concept description.

Whereas holistic descriptions pay attention to all relevant aspects of a given topic, their very nature cannot include the background and the practical aspects of One Health implementation in specific diseases. It is one thing to have a One Health conceptual framework and another to apply it, particularly when we consider that many essential elements of this framework overlap (Figure 1), making it difficult to establish priorities. In brucellosis, for example, awareness of the disease is necessary and challenging because of its protean characteristics [3]. In addition, zoonosis awareness depends on Animal and Public Health cooperation and ancillary notification and surveillance, which depend on the correct use of diagnostic tools [3]. These are implemented when there is awareness, and they require both know-how and technical infrastructure, aspects that should be optimized by Animal and Public Health cooperation. Therefore, these interactions are multidimensional and the order in which they are discussed below does not represent their priority.

## 3. An Unmet Challenge: Implementing Basic One Health Structures

At a global level, no studies have systematically analyzed the extent of effective collaboration between Animal and Human Health sectors and ancillary surveillance, infrastructure, plans for intervention, legislation, and other measures of countries affected by brucellosis. However, the WHO, FAO, and WOAH have launched programs to assess the degree of implementation of some of these elements as a necessary step to strengthen implementation in countries throughout the world and, although not objective (participation is voluntary and data are provided by the corresponding administrative institutions/agencies of each country), the reports provide some information [24,25,26].

The WHO Joint External Evaluation (JEE) reports are voluntary, collaborative processes to evaluate the capacity to prevent, detect and rapidly respond to public health risks. For prioritized zoonoses, the method uses scoring tables and specific questions on critical points regarding functional capacities in the animal health and public health sectors and the existence of effective collaboration, coordination and communication. The results are then reflected as quantitative scores [24]. Recently, Elton et al. [26] reviewed the JEEs of 44 sub-Saharan African (SSA) nations, and the relevant conclusions on zoonotic diseases are summarized in Figure 2 and Table 1. While there is no specific information on brucellosis, it was prioritized by 34 of these 44 countries; therefore, these analyses approximately apply to this zoonosis.

As can be seen, most SSA countries are deficient in their implementation of the collaborative systems essential to the application of One Health in the control of prioritized zoonoses. Considering that JEEs are voluntary self-evaluations, an obvious conclusion is that these SSA are mindful that implementing these requisites will be an immediate challenge for the years to come. These concerns are confirmed by specific studies on One Health’s implementation in Ethiopia, Kenya and Uganda that include brucellosis and emphasize deficiencies in data-sharing and communication (and the implicit notification systems), and the need for continuous advocacy among the community for financial funds and technical capacity [17,20,21].

The WOAH provides a tool to evaluate the Performance of Veterinary Services (PVS), including the ability of veterinarian authorities to coordinate with other bodies that play a role in One Health [25] and, according to Stratton et al. [27] about 140 countries had joined the program by 2019 (the tool was updated in 2022 [28]). The evaluations are voluntary and the reports are confidential. However, some countries have authorized the WOAH to make the reports public, and the 2019 PVS webpage contains 12 for Ibero-America, 6 for Asia, and 13 for Africa. These reports are divided into four sections: Human, Physical and Financial Resources, Technical Authority and Capability, Interaction with Interested Parties and Access to Markets, each with precise points/questions with a score from 1 to 5 (5 being optimal).

For the first three sections, we used the scores to obtain averages for countries not covered in the JEE reports (Table 2), even though there are differences between countries in the number of specific points in each section.

Although parallels between these figures and the ability to implement One Health policies should be established cautiously, comparisons with Canada and Japan point out deficiencies in scores below 4. This idea is supported by the score of India in Table 2 because it coincides with the assessments made in a recent collaborative workshop on One Health and brucellosis focused on this country [19]. There are no publicly available PVS reports for Mexico, Perú, Colombia, Ecuador, Iran, several countries in central Asia, Pakistan and China, which are all heavily afflicted by brucellosis.

For Iberian America, some information on these issues can also be obtained from a survey conducted under the guidance of PANAFTOSA/WHO, which included 12 South American and 5 Central American countries, Mexico, and 15 Caribbean countries [29]. However, a country-by-country interpretation is impossible because the data published are a pool of answers from very different countries, and specific results for each country are not currently accessible. This lack is relevant because brucellosis in Mexico, for example, has a different dimension from most Caribbean nations, and the latter are quantitatively overrepresented in the average, thus biasing the data. Despite this bias, some results are worthy of note.

The report shows that the Ministries of Health and Agriculture declare a marked disparity in priorities concerning brucellosis, with the former declaring brucellosis a lower priority. It is unclear whether this is due to the use of different prioritization criteria by the respective Ministries, lack of coordination, limited diagnosis, a deficient notification, the implementation of milk pasteurization in some of these countries (which would reduce brucellosis’s impact on humans) or a combination of factors. The criteria most frequently reported as being used by the Ministries of Health in their prioritization were, in order of frequency, human disease incidence, human disease severity, human disease mortality, and human disease prevalence. For the Ministries of Agriculture, the criteria were economic impact, animal disease prevalence, human disease incidence, and animal disease incidence. Remarkably, this survey shows that the Ministries involved do not contemplate control and coordination as capacities that require improvement, and diagnosis, education and surveillance also did not score highly. For example, only 12 and 6% of the answers considered that control and coordination, respectively, require improvement. This judgment suggests a degree of satisfaction with control and coordination in official instances (answers were primarily provided by zoonosis program managers within each Ministry) that is not proportionate with either the endemicity of brucellosis or the little progress in control achieved in most of the countries included in this PANAFTOSA/WHO survey.

In addition, One Health studies cover the same issues in some endemic countries in other areas. In the Middle East, a study in Jordan stressed that it is essential to review the reporting and surveillance systems and to develop capacity-building in zoonotic disease diagnosis [15]. In Asia, a collaborative workshop on One Health and brucellosis in India held in 2017 included transboundary and transdisciplinary collaboration as priorities [19], and these problems are implicit in the major obstacles identified in neighboring Pakistan [30]. Noteworthy, brucellosis is endemic in humans and animals in both countries, and India has one of the largest ruminant populations and is the largest milk-producer in the world. To our knowledge, studies like these do not exist in other parts of the world.

Finally, fragile states are defined as those that can no longer perform essential functions such as education, security, or governance because of violence or extreme poverty [31]. We could consider that states with a high warning, alert, high alert or very high alert score in the fragile state index (54 countries in 2022; [31]) are unlikely to have a functional infrastructure that can tackle zoonotic diseases because of extreme poverty or violence.

## 4. One-Health Challenges for Brucellosis

Below, we will discuss challenges that are specific to brucellosis. As said, the order of discussion does not imply priority.

### 4.1. Awareness of the Disease

Awareness is essential to implement One Health programs against any disease. In animal brucellosis, lack of knowledge about the disease results in constant economic losses and its perpetuation in humans, underdiagnosis, and subsequent suffering, or a delay that increases the risk of complications and therapeutic failure/relapse [32]. However, awareness is particularly problematic in brucellosis because of the lack of specific symptoms and signs in humans and animals. Moreover, signs are seldom noticed in herds/flocks under extensive breeding, a production system typical of many endemic areas (see below).

These issues are confirmed in a recent review covering 79 studies in 22 countries, most from Africa (24 studies) and Asia (49 studies), published in several languages (including Chinese) and focused on high-risk groups (animal and human health workers, livestock owners, dairy farmers, abattoir workers, pastoralists, patients, students and residents [33]. It is worth noting that the review comprises studies in China and India, which, with 2.8 billion inhabitants, contain one-third of the world population [34] and are home to about 400 million cattle, 150 million buffaloes, 262 million sheep and 171 million goats, representing 40%, 86%, 60% and 15%, respectively, of the total world populations [35,36].

The pooled awareness levels regarding the zoonotic nature, mode of transmission, and signs of human and animal brucellosis are presented in Table 3. As shown, the highest awareness level (human brucellosis signs) only scored 41%, and awareness about the existence of animal vaccines was the lowest. The meta-analysis authors did not find significant differences between high-risk populations in Asia and Africa for all investigated topics, but there were differences in the results for vaccines (15 studies), with pooled awareness levels of 4% and 46.3% in the African and Asian populations, respectively. The conclusion is obvious: insufficient awareness and knowledge of brucellosis are widespread in at least the largest countries of Asia and most African nations. This is unsurprising since many suffer from weaknesses in Public Health and Veterinary Services and notification systems. Few studies in this meta-analysis cover other areas where breeding susceptible livestock is also very important and brucellosis exists, but the conclusions about Asia and Africa are likely to apply elsewhere.

### 4.2. Geography, Climate, and Peri-Urban Growth

Small ruminants, cattle, and possibly water buffaloes are the most numerically important hosts of zoonotic brucellae and, in contrast to pigs, they are bred under conditions that often depend on geography and climate. Table 4 summarizes their distribution according to production systems. Although the data were gathered about 15 years ago [37], they illustrate several key points. First, the majority of these animals are in resource-limited countries. Second, vast numbers of these animals are in grazing and rainfed-mixed production systems (i.e., extensive breeding systems), and when the numbers in Table 4 for these two categories are compared with the map in Figure 3, it is evident that the co-grazing of two or more ruminant species is extended. As expected, in large areas of the world, grazing and rainfed-mixed systems often border each other (Figure 3). The well-known implications of these circumstances are animal movements in large, often difficult-to-access areas, transhumance and, for some parts of the world, ethnical and political conflicts, as are notorious in countries bordering the African Sahel [38]. These data illustrate the enormous challenge posed by any assessment of brucellosis prevalence and its control and eradication where most susceptible livestock reside, which are all essential components of One Health approaches.

In addition, other challenges derive from the distribution of *Brucella* hosts. These figures and geographical distribution are not static. Climate and demographic changes and demand for livestock are behind the rapid changes in ecosystems, ecosystem intrusions/invasions, and people’s movements, the three main proximate drivers of changes in livestock disease dynamics [39]. Climate change projections show disproportionate effects in Africa. It has been estimated that warming and drying may reduce crop yields by 10–20% in the next 25 years in Africa and, in this context, livestock may provide an alternative to cropping, particularly in areas where crop production is already marginal [40]. Indeed, not only would draught-resistant ruminant breeds be favored, but also camels, animals affected by brucellosis about which there is almost no information on diagnostic tests and none on vaccines (see below). Many such areas are in the Sahel, overlapping with areas of high cattle density (grazing and/or rainfed–mixed breeding) (Figure 3), and there is a risk that nomadic or semi-nomadic herds where brucellosis is endemic are pushed south, spreading the disease into rapidly growing peri-urban areas facilitated by the cattle corridors that merge in the big cities [41]. In almost all African countries, peri-urban interfaces are places with a high risk of disaster outbreak because their rapid growth is not paralleled by an appropriate infrastructure [42,43] and infected herds have the potential to trigger severe brucellosis outbreaks in animal and humans, as previously observed in Nigeria [44].

### 4.3. Intensification of Breeding

Table 4 shows industrial breeding (meat and/or milk) in industrialized countries (landless production systems in Figure 3). These systems need prudent management because, due to the increasing stock density and movements of animals, people and vehicles on and off farms, intensification favors the spread of zoonosis [45], a well-known fact in brucellosis. In 2019, large dairy companies in China had 1.7 million dairy cattle or, on average, nearly 68,000 animals, each with milking parlor units for 15,000 animals [46], and some farms contained about 100,000 animals [47]. Herd size is a known risk factor for brucellosis and when the number of animals is enormous, it becomes exceedingly difficult to control because the exposure to *Brucella* is increased above the protection level provided by S19, the best cattle vaccine [48]. Not surprisingly, a recent review reports that the individual seroprevalence in dairy cattle herds in China raised from 1.6% in 2008–2012 to 2.6% in 2013–2018, and, in some areas, is higher than 30 or even 50% [49], even though 66 of the 88 reviewed articles used the tube serum agglutination test (SAT), which is less sensitive in cattle [50].

### 4.4. The overlooked Characteristics of Brucellosis

To the challenges imposed by geography, climate and intensification, we must add the negative impact of several extended misconceptions concerning animal and human brucellosis, diagnostic tests and vaccines. They contribute to the lack of reliable evidence for governments and policy-makers to act on at both local and national levels, even when applicable legislation has been approved, which is not always the case because, in a kind of vicious circle, legislation and planning await “evidence” (see below).

There is an increasing number of publications aimed at determining animal seroprevalences and risk factors in areas where the disease is known to exist through abattoir studies, reports of human brucellosis in risk groups and other evidence. However, these studies often have design problems because of difficulties related to the characteristics of this disease (Table 5). These problems lead to incorrect assessments being made before any control measure is implemented and prevent a correct estimation of their effect. However, both data sets are essential in order to not frustrate One Health approaches.

It has been known for almost 80 years [51] that a chief characteristic of this disease in domestic ruminants is that, in a given herd or flock, it can fluctuate between a suddenly noticeable presentation with abortion storms to a state in which, although the infection is permanent, this symptom is hardly evident, with the latter being the most common situation in extensive breeding systems. The experience of Spain and other European Mediterranean countries is that, although individual prevalence was moderate in many endemic areas, the high number of infected herds/flocks indicated a high risk of rapid brucellosis expansion and posed a severe barrier to control. Another relevant characteristic of the disease in ruminants is that, no matter the situation, the disease primarily affects animals of reproductive age. Overlooking these facts results in imperfect assessments. When they lead to the false conclusion that the overall individual prevalence is low, it is commonly assumed that brucellosis does not represent a significant threat and can be controlled by test-and-slaughter, a critical mistake [3]. Illustrative of these problems, in a relatively recent review of brucellosis in Sub-Saharan Africa (SSA) [52], only 20 out of 34 publications published in 2002–2017 reported herd/flock seroprevalence. In addition, in many of these 34 studies, a random selection of animals included the epidemiologically irrelevant young ones, thus reducing the probability of detecting infected herds/flocks. One or more of these problems are also common in more recent works, thus limiting the value of the studies [53,54,55,56,57,58,59,60,61,62,63,64,65,66,67,68,69,70,71,72].

Similarly overlooked is the significance of cross-infections (Table 5). For instance, most studies investigate only one species, usually cattle, in areas where the existence of other livestock and the particular breeding system strongly suggests that they are bred together or in contact with other ruminants [52,53,55,57,65,66,67,68,73,74,75,76]. Regrettably, this is accompanied by the assumption that *B. melitensis* cannot infect bovines or that, if this happens, cattle spontaneously clear *B. melitensis* once contact with infected small ruminants is prevented [77]. These misconceptions are repeatedly proved wrong in Mediterranean countries [48,78,79,80,81,82,83].

### 4.5. The Negative Impact of Diagnostic Test Misuse

Correct assessments also require the appropriate use of tools, and issues that often negatively impact the diagnosis of animal brucellosis are summarized in Table 6.

It is of paramount importance [50] to standardize the antigens and other components of the tests. The smooth-to-rough dissociation is a real issue in the preparation of antigens for the diagnosis of all classical smooth *Brucella* species, and titrating them and monitoring their quality is a strict necessity [84,85]. Few reports have addressed this point in the endemic countries that produce their antigens, but one study in Iran reveals the importance of this issue [86]. For quantitative tests, it is also necessary to establish the cut-offs using sera that are representative of the area to circumvent specificity issues related to the variable background reactivity of sera of animals raised under very different conditions [50]. When provided, it is not correct to rely on the cut-off in manufacturer instructions, and these often suggest a relatively sizeable cut-off span of little practical use. Adopting indirect ELISAs (iELISAs) that are not locally validated as the primary test is common. This adoption possibly results from the direct transposition of the use of some of these ELISAs for surveillance in brucellosis-free countries that have well-equipped laboratories, proficient animal identification systems, easy access to animals that allows for repeated sampling, and high manpower costs that justify investments in appropriate equipment [3], circumstances that are not those of many endemic countries. These issues also affect competitive ELISA (cELISA) and the Fluorescence Polarization Assay (FPA) [87].

In summary, the efficient use of these relatively sophisticated tests requires a proper understanding of the tests themselves, laboratory infrastructure problems to be resolved, animal tagging, repeated access to animals, and other issues. In any case, these assays do not outperform the diagnostic sensitivity and specificity (DSe and DSp) of the classical buffered *Brucella* agglutination tests (Rose Bengal [RBT], Card and Buffered Plate Agglutination tests) that do not suffer from cut-off adjustment problems [50]. There are abundant examples of these problems in the recent literature [52,53,54,55,56,61,62,66,68,71,88,89,90,91,92,93,94,95,96,97,98,99,100,101,102,103,104,105,106,107,108].

A common mistake is the use of a “confirmatory test strategy”, typically combining the RBT for “screening” with either the complement fixation test or iELISA or cELISAs (or SAT, which is poorly sensitive) as “confirmatory” tests. For example, in a 2016 review of brucellosis in SSA, 17/34 studies used this incorrect strategy [52], and the problem persists in this subcontinent and elsewhere [57,58,59,63,67,68,69,73,74,75,76,109,110,111,112,113,114,115,116,117,118,119,120,121,122,123,124,125]. Provided it is properly understood and implemented, this serial RBT/complement fixation strategy can help to minimize the number of animals culled whenever vaccination with S strains (S19 or Rev1) is part of a correctly implemented combined eradication program in which it is feasible to conduct a repeated testing of RBT-positive animals to follow up titer decreases/increases in complement fixation [3]. Otherwise, it increases costs and produces incorrect results. This problem can also be found in assessments of domestic pigs [126].

A significant number of studies misuse the milk-ring test. Although this test has only been validated in the milk of cows (*Bos taurus*) and proved to be no good for small ruminants [127], it is often used in sheep, goat, water buffalo and camel milk [52,60,128,129]. No studies have validated the milk-ring test in these animals and as it depends on the specific characteristics and content of the fat in cow milk, it cannot be assumed that it works in other ruminants and in camelids [127]. Some iELISAs that have not been validated for diagnosis in milk are also often used to test this sample [60,105,128,130,131].

Authors, editors and reviewers often overlook these issues, and most reviews and meta-analyses of seroprevalence in endemic areas use data without critically evaluating the tests in the original reports, thus spreading misinformation. Examples include reviews and meta-analysis aiming to assess seroprevalence and risk factors in Nigeria [132], Cameroon [133], Tanzania [134], Ethiopia [135,136], East Africa [137], the whole Africa [138,139], the Middle-East [140], Mosul (Iraq) [141], North Eastern India [142], India [143], China [22,49,144,145], Asia [139] and the whole world [146,147].

Molecular methods represent a significant advance in the reliable identification, typing, and genotype analysis of *Brucella* strains previously isolated by culture. PCRs have also been applied to detect *Brucella* DNA in animal tissue or blood samples. These are very attractive tests because they have the potential to provide superb DSe/DSp and simultaneous species identification. Nevertheless, the absence of internationally agreed protocols, the variety of DNA extraction and purification protocols, primers, lack of standardization, and proper validation are unsolved issues [3]. It is not uncommonly assumed that analytical sensitivity and specificity equate to DSe/DSp. However, analytical sensitivity does not evaluate potential inhibitors and extraction issues in natural samples. In addition, testing the analytical specificity with known potential “cross-reactors” (if carried out) assumes that these represent, in variety and numbers, the microbiota that could cause inespecificity in biological samples and does not address the problem of the “unknown unknowns”. A recent review identified 73 studies in domestic ruminants, camels, and pigs, 70 of which are flawed regarding the validation of the DSe/DSp of the used protocol [3].

Implicit in these problems is the need for reference laboratories (and capacity building) in key areas of endemic countries. It is symptomatic that most official FAO/WOAH brucellosis reference centers are found in countries from which brucellosis was eradicated decades ago, that there are none in countries in SSA, Central Asia and the Indian subcontinent, and only one in Iberian America [148].

### 4.6. Human Brucellosis Prevalence

There are no reliable records of the global prevalence of human brucellosis and the constantly repeated 500,000 annual cases figure lacks any support [3,4]. Based on 29 scattered studies that were judged to be of sufficient quality for data analysis [149], a WHO global estimate suggests that the number of human cases could be between 340,000 and 19,500,000 in 2010, half of which could be foodborne [150]. This scarcity of reliable prevalence studies and the exceedingly high range of this estimate illustrates the absence of data. As discussed below, documenting the actual prevalence of human brucellosis is even more decisive in sensitization than similar data on animal disease, and it is a crucial element of One Health.

The interrelation between awareness and diagnosis is critical. This fact is exemplified by the events in China, one of the world’s most populated countries and home to many brucellosis-susceptible animals in all types of husbandry (see above). According to Lai et al. [151], the number of human cases reported between 1955 and 2003 in this country did not vary significantly, most often below a few thousand and seldom reaching 10,000 per year. However, the figures rose to about 57,000 in 2014 [152], and then increased steadily but not markedly [153]. The incidence of other zoonotic bacterial infections such as anthrax and leptospirosis decreased in the same period. This decrease suggests that improvements in animal and human health services and the ensuing increase in awareness and diagnosis are one factor behind the rise in brucellosis prevalence figures.

In addition to a keen awareness of the possibility of contagion, the diagnosis of human brucellosis requires appropriate laboratory tests because the disease lacks specific signs and symptoms [3]. Since they need to be included in a proper brucellosis case definition, understanding laboratory tests is essential from a medical standpoint to obtain prevalence figures that are not distorted by misdiagnosis and underreporting. The last update of the brucellosis case definition of the USA Center for Diseases Control and Prevention (CDC) [154] considers that, in the absence of a positive *Brucella* culture (significantly, this culture is not feasible in large areas of the world), definitive evidence for brucellosis is “a fourfold or greater rise in *Brucella* antibody titer between acute- and convalescent-phase serum specimens obtained greater than or equal to 2 weeks apart”. (sic.) According to the same source, presumptive evidence is a “*Brucella* total antibody titer of greater than or equal to 160 by standard tube agglutination test (SAT) or *Brucella* microagglutination test (BMAT) in one or more serum specimens obtained after the onset of symptoms” or “Detection of *Brucella* DNA in a clinical specimen by PCR assay” (sic.). However, for culture, these definitions are imprecise and, if used as a reference when evaluating diagnostic tests in endemic areas, are a source of biases, as the recent literature shows [155]. Titers and their variations depend on the test (for example, they vary widely between SAT or BMAT, on the one hand, and Coombs or Brucellacapt, on the other, to cite commonly used tests), seldom rise after the first diagnosis, not during treatment and convalescence, unless a relapse occurs. They drop in most patients for agglutination tests, and drop more in SAT and BMAT, and IgM, IgA and IgG ELISAs [156]. Concerning SAT and BMAT, titers are usually lower than 160 or even negative in long evolution cases (see, for example, [157]) because these tests do not detect non-agglutinating antibodies, which are characteristic of the long evolution cases common in endemic areas with no immediate access to health care [158]. Finally, since there are no standard PCR protocols for direct diagnosis, it cannot be assumed that any PCR provides adequate DSe and DSp [159]. Correct case definitions are crucial in assessing human prevalence because they serve as templates in countries with little clinical and laboratory experience in this disease that are beginning to become aware of its importance.

Concerning tests, recent studies revealed an issue of unsuspected importance. “Febrile antigen” kits for human brucellosis are marketed and widely used in low- and middle-income countries in Africa and Ibero-America and, as far as we have documented, in Pakistan. Studies in Kenya and Tanzania [155,160,161,162,163,164] have conclusively shown that these tests are a source of a very high proportion of false-positive results (over 95% in 800 febrile patients [160]), in all likelihood because of the smooth to rough dissociation issues [3]. Their use causes unnecessarily prolonged, expensive treatments and makes many records of human brucellosis prevalence unreliable, as found in Tanzania [162]. Even worse, the disconnection of the epidemiological data and true brucellosis hides the effect of any control measures (hygiene, vaccination, and other issues), generating skepticism and mistrust of One Health recommendations among stakeholders, particularly farmers (see below).

### 4.7. Brucellosis Vaccines: Efficacy Versus the DIVA Myth

No human vaccines are available, and animal vaccination is essential to control the disease and reduce its impact on humans in areas where brucellosis is endemic. Although other vaccines were used or are used in some countries where the disease is still endemic [3], there is only scientifically valid information on the three WOAH-recognized ones, two for cattle (S19 and RB51) and one for small ruminants (Rev 1) [48]. The critical issues in the use of Rev 1 are discussed below, and Table 7 summarizes the properties of S19 and RB51 that should be considered when selecting a vaccine for use under the conditions prevailing in endemic areas (see Table 4 and Figure 3) or in infected herds elsewhere.

S19 has been used to eradicate brucellosis in many countries, including the United States, Australia, Canada, New Zealand, and several European countries, including Spain [165], where *B. melitensis* infections of cattle in some areas were very significant [48]. This fact disproves another misconception: that the etiological characterization of the *Brucella* species provides essential information guiding vaccine choice [166]. Success has not accompanied the use of RB51, even when repeated doses are administered, considerably increasing costs [48]. This procedure (never tested under controlled conditions) was introduced almost three decades ago for unknown reasons, possibly due to the suspicion of insufficient and/or rapid loss of protective efficacy, as recently confirmed under controlled experimental conditions [167]. RB51 has serious safety issues in animals in addition to two critically important ones in humans: infections cannot be diagnosed using current serological tests and the vaccine is resistant to rifampin, an antibiotic that is part of a combined regime to treat brucellosis [168,169,170].

Interference in serological testing is repeatedly cited as a paramount inconvenience of S19, and has become an absolute dogma advising its replacement by RB51. Considering that most countries where RB51 is marketed cannot apply a test-and-slaughter policy, the reasons behind this insistence are unclear [3]. Nevertheless, it has been so persistent and extended that only RB51 is marketed in many countries. This allegation is not grounded in reality [48]. First, the S19 “problem” is chiefly solved by conjunctival administration during calfhood and the proper serological follow-up of vaccinated herds. Second, the serological follow-up of vaccinated herds is impossible/irrelevant wherever access to the animals, tagging, and repeated testing is not possible, or where the removal of suspicious animals (for example, in test and slaughter policies) is not feasible, including the lack of funds that are common in low- and middle-income countries. Third, while it is true that RB51 is a rough strain and is thus considered a DIVA vaccine, a crucial question is to what extent the lack of the key diagnostic O-chain epitopes is useful in areas where vaccination is necessary. Because of the high immunogenicity of the smooth lipopolysaccharide, an RB51-vaccinated animal will develop antibodies to the O-polysaccharide upon exposure to virulent brucellae. This simple and unavoidable fact abrogates the practical use of the RB51 DIVA properties in tests such as RBT or complement fixation in endemic areas. Furthermore, either with or without contact with wild-type brucellae, RB51 interferes in ELISAs, cELISAs and LFiC, and possibly in FPA, because of the cross-reactivity of the rough and smooth lipopolysaccharide in immunosorbent assays [127]. None of these tests would differentiate infected and vaccinated exposed animals in endemic areas where RB51 is implemented (on the assumption that vaccinated animals remain seronegative). This delusion results in confusion and over-culling if combined with test-and-slaughter eradication policies, fueling mistrust (see below) in farmers [3].

The quality control of vaccines is of paramount importance. The genetic drift of brucellae upon subculture is a well-known problem, of which the smooth to rough dissociation is only one example, which is relatively easy to monitor. However, the correct attenuation/protection balance can only be controlled in a biological model, and this is essential to reproduce the properties of the master seed in seed lots. There is an OIE-approved protocol for S19 to control residual virulence and immunogenicity in mice but the quality control of RB51 is only carried out in vitro by plate counts of viable bacteria, an insufficient criterion [85]. Therefore, it is unknown how the properties of the original strain and batch-to-batch uniformity have been reproduced since RB51 was introduced over 30 years ago.

In summary, all existing evidence is against the widely accepted idea that RB51 is advantageous over S19. Its use in many countries is rather detrimental in essential components of One Health strategies: in control, it has not proved its efficacy wherever the impact of cattle brucellosis by either *B. abortus* or *B. melitensis* is essential; in serodiagnosis, it may be a source of confusion.

### 4.8. Capacity Building

Overall, there are initiatives to strengthen capacities in less favored countries. The JEEs and PVS reports discussed above help to identify needs, and the latter includes missions that recognize gaps and address sustainable laboratory support. Still, the PVS web’s latest update (2020) shows reports of only five countries [28]: Bhutan, Costa Rica, Côte D’Ivoire, Kyrgyzstan, and Sudan. They contain general information about the analyzed samples/diseases, workflow, cost analysis, and other logistic issues. However, they do not address specific concerns about training in brucellosis, the performance of tests, and other relevant practical matters.

As mentioned, there are no WOAH/FAO brucellosis reference laboratories in SSA, Central Asia and India, and only one in Ibero America. In contrast, there are four in Western Europe, three of which are found in countries where brucellosis was eradicated long ago. To bridge this gap, the OIE PVS has a Laboratory Twinning Program to link “parent” and “candidate” laboratories. Table 8 shows the brucellosis projects completed in December 2021. Obviously, the number of “candidate” laboratories compares unfavorably with the above data on the extended deficiency in Veterinary Services in Africa and other parts of the world (Table 1 and Table 2). Also, none of the countries in Table 8 have become home to a WOAH Reference Laboratory for brucellosis, except Thailand and the United Arab Emirates.

These efforts are not limited to the PVS programs. The literature shows many publications co-authored by brucellosis workers from, for example, Sweden, Germany, the UK, Switzerland and Spain, in endemic countries, such as Egypt, Pakistan, Kyrgyzstan Kenya, Tanzania, Uganda, Morocco, Argelia, and Mongolia. Inevitably, most “parent” laboratories belong to the limited set of countries that eradicated brucellosis from cattle and small ruminants with means and conditions differing from most of the “candidate” laboratories. Because of this, in most countries of the parent laboratories [148], know-how based on direct experience with real brucellosis is waning and the priorities are surveillance, false-positive serological reactions and other issues not relevant in endemic countries. This issue risks generating a gap that may create an obstacle to effective capacity-building that is applicable in the conditions of the countries where it is needed.

### 4.9. Factual Technical Gaps in Diagnostics and Vaccines

We list here what are, in all likelihood, the most critical gaps.

*A safer small ruminant vaccine*. Mass-vaccination (i.e., whole herd/flock) is required to control animal brucellosis in most endemic countries [3,48,171]. It is well-known that vaccine Rev 1 is highly abortifacient, thus limiting its use in mass-vaccination. Despite this, Rev 1 was instrumental in the eradication of small ruminant brucellosis in, for example, France and Spain because of the existence of efficient Veterinary Services, sufficient farmer awareness, and the lambing/kidding period in Mediterranean countries, which esablishes a safe window of time to minimize the vaccination of pregnant animals. Wherever such a window does not exist (a common situation in many resource-limited countries), the Rev 1 abortifacient effects represent a critical obstacle, which is practically insurmountable when mass vaccination is necessary [171], making a safer small ruminant vaccine a crucial need. Developing a safe small ruminant vaccine is the target of the GalvMed Brucellosis vaccine competition (https://brucellosisvaccine.org; accessed on 6 May 2023). Nevertheless, another challenge concerns the necessary experiments in the target host under controlled conditions. Due to the current regulatory rules, these experiments need an exceedingly expensive P3 infrastructure that is not widely available. Moreover, these studies must fulfill a series of critical technical requisites concerning brucellosis-free animals, protocols regarding the challenge and sensitive detection of infection, and controls that have been overlooked since the correct experimental conditions were established over 30 years ago (reviewed in [3]).

*Diagnostic tests and vaccines for water buffaloes, camelids, and yacks*. Brucellosis in these animals has been hardly investigated and, despite extended assumptions, there is no reliable information on the validity of serological tests or the safety and protective efficacy of current vaccines.

*Brucellosis in wildlife*. Regarding anthropogenic factors, in the best-known cases [172,173], the impact of brucellosis in wildlife is challenging to assess because the only specific diagnostic tool is bacteriology, which is difficult to implement for obvious reasons. The serological tests used in domestic animals await validation in wildlife species, and the direct application of PCR to field samples has the same DSe/DSe uncertainties as in domestic animals [3,174]. It is worth noting that the countries that eradicated the disease from the latter animals did not include any sanitary interventions in wildlife [175], showing that, in these countries, brucellosis in wildlife is of little or no epidemiological relevance to the disease in domestic livestock.

### 4.10. Taxonomy Issues

The concept of brucellosis needs a clear taxonomical status for the pathogen and a proper nomenclature. From the One Health perspective, the proposal to include the phylogenetically related opportunistic Ochrobactrum in the *Brucella* genus could cause severe harm and confusion [176]. There are also solid taxonomical arguments supporting the idea that Ochrobactrum organisms are very different from *Brucella* organisms and, on practcal grounds, these two bacteria must be kept apart in different genera [177], their current merging in data bases must be corrected, and the growing and harmful confusion in the recent literature should be avoided [178].

## 5. The Human Factor

### 5.1. Building Trust in Key Brucellosis Stakeholders

In a recent analysis of brucellosis control (mostly in sheep) in the Negev, Hermesh et al. [179] point out that trust in the Public and Veterinary and Human Health systems is essential in One Health approaches to combat brucellosis. In Israel, an increasing incidence of *B. melitensis* has been reported in recent years, up to 100-fold higher in the Bedouin Arab population than the Jewish population [180]. While the reasons for this disparity are multiple, the study identified two entwined reasons why Bedouins refused to cooperate in brucellosis control, grounded in mistrust. Bedouins felt that the government wanted to eliminate the flocks to drive them out of the desert into towns and obtain better political control of them. Second, as indicated above, the farmer’s involvement also hangs on the realization of the impact of this disease on their animals, which is likely to be clear in intensive but not extensive breeding conditions, as well as the effect of infection in them and their families. Hermesh et al., [179] recorded this very explanatory statement:


*Sometimes, you tell a herder to boil the milk, and he’ll reply “my sheep are fine, I just checked them,” …. “My father and my grandfather never boiled, and they were just fine”.*


This attitude leads to this complaint by veterinary personnel:


*When MoA veterinarians come to cull animals, the Bedouins conceal them, transfer them to other owners, or claim they were stolen.*


Needless to say, this is not specific to Bedouins, although these political issues accentuate mistrust. In the authors’ experience, this mistrust was also relevant in Spain, Italy and Ibero-America, as it is almost everywhere. In addition to Public Health and Veterinary Services, microbiologists, medical and veterinary practitioners, One Health’s implementation in brucellosis strictly requires the active involvement of breeders, and experience demonstrates that failure to do so is an almost unsurmountable obstacle to control and eradication [171,181]. If programs count on the “willingness to pay” of breeders who are not affluent, it is not difficult to predict what will happen. Indeed, no country has eradicated brucellosis using programs paid for, either totally or partially, by the animal owners.

Building trust should be one critical goal of One Health programs; thus, it should exist at all levels, including between the Veterinary and Human Health systems. Organizations are made of individuals, who must establish trust. In the experience of two the authors of this paper, many physicians had little or no trust in Veterinary Services as far as brucellosis was concerned in the years in which human brucellosis was almost a public scandal in Spain.

These issues merge with two general ethical principles: social justice and political peace [18,182]. Remarkably, they were implicit in Schwabe’s statement that One Medicine (One Health) must face the “immensely complex, multifaceted future quality of human life and, ultimately, of human survival”.

### 5.2. Prioritization Versus Sensitization

A scan of the accessible evidence shows that brucellosis is very often ranked among the prioritized zoonoses in which these analyses were conducted [5,19,183,184,185]. Therefore, establishing brucellosis as a priority does not seem to be a key issue. However, establishing aspects related to human conduct could be more important and elusive than only presenting the facts. They are implicit in the statement of Delia Grace, a person with extensive experience with zoonoses and their impact on the poorest communities in the world. In an interview [186], she stated:


*What will make politicians change their minds? That is an interesting area but it may be that simple things like evidence or financial information is not very compelling. So, what should we do?*


The experience of most European Mediterranean countries is that the dimension of human brucellosis led to a push for control and eradication. For example, in Italy, the number of human cases peaked at 9538 declared cases in 1950, and in Spain there were nearly 8000 declared human cases (the actual figure could be twice as much) in 1980. Expressions such as “excruciating pain” “permanent disability”, “muscular impotence”, “wretched appearance”, “drawn facies” “listlessness”, “utter state of exhaustion” and “pathetic discouragement” were used by Spink in his classical monograph to describe the disease [187]. They illustrate why the human side of this zoonosis is so compelling, more than simple evidence of the presence of the disease or financial information. However, in many middle-income countries where *B. melitensis* is absent and pasteurization is the rule, brucellosis, mainly caused by *B. abortus*, does not have as dramatic a zoonotic impact as in other countries where the former species is the leading cause of human suffering; therefore, governmental authorities tend to overlook the problem [188]. Nevertheless, the medical profession has played a crucial role in revealing the intolerably high number of cases worldwide, some of which may be significant. There are examples of brucellosis affecting family members of relevant politicians that coincide with renewed efforts of control. Whether this represents a fortuitous coincidence or owes something to a vigorous sensitization of policy-makers to the “facts” is left to the reader’s interpretation.

## 6. Conclusions

The main challenges in applying One Health approaches in brucellosis are not the adoption of the concept itself, as the history of the disease proves. Instead, they relate to a series of issues, both general and specific. Some are inherited; the fast-evolving political and socio-economic factors and climate changes create others. They can be listed as the specific sections discussed here:A.The deficiencies in Public Health and Veterinary Services and their cooperation.B.Insufficient awareness.C.The challenges related to geography and climate.D.The intensification of issues to meet food demands.E.Misconceptions about the disease, diagnostic tools and vaccines.F.Research on diagnostics and vaccines for water buffaloes, camels, yacks, etc.G.Capacity-building.H.The human condition: care for others, building trust and social justice.

Whether all can be tackled simultaneously is a question with a noticeably negative answer. The most important is the last in the list, but it would require a utopia, and is not something that, beyond personal actions, scientists can fix. Others, such as population growth, global warming, erosion, and environmental damage, cannot be controlled by individual actions. These uncertainties leave three tasks for professionals: E, F and G. They are not small challenges; it would be an achievement if efforts could be strengthened and coordinated. Following the title of a 1902 book by Piotr Kropotkin, the anarchist that opposed radical forms of social Darwinism: “Mutual aid: a path of evolution” [189].

## Figures and Tables

**Figure 1 microorganisms-11-02070-f001:**
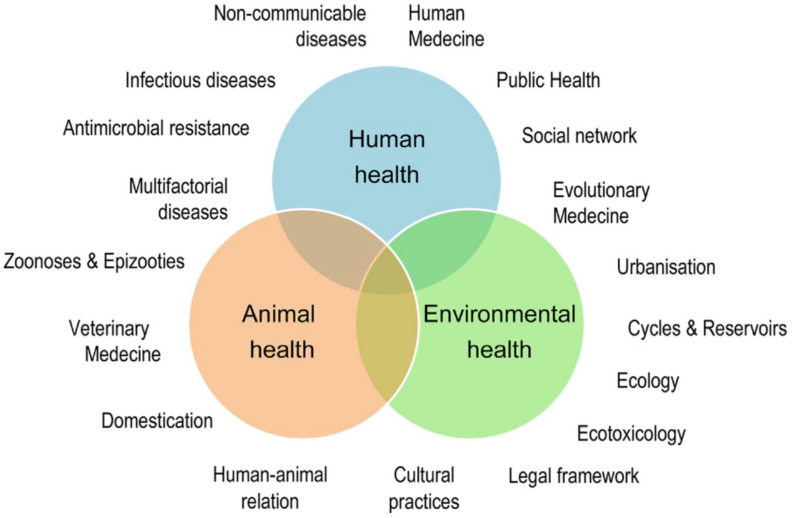
The One Health concept: a holistic, transdisciplinary, and multisectoral approach to health (adapted from [23]; reproduced with permission).

**Figure 2 microorganisms-11-02070-f002:**
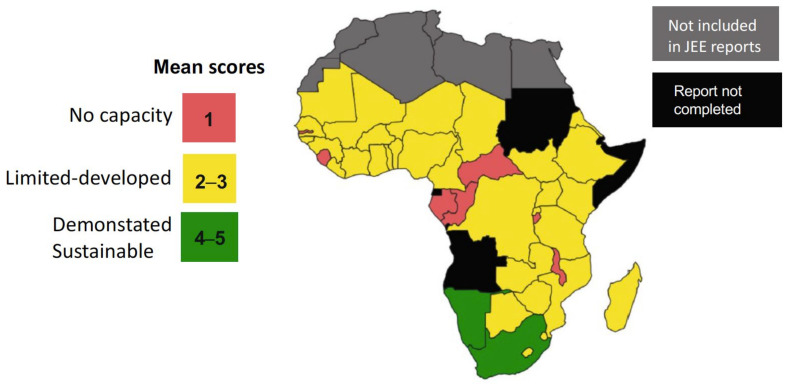
Map showing SSA country mean zoonotic disease preparedness scores (modified from [26]; reproduced under the Creative Commons Attribution 4.0 International License).

**Figure 3 microorganisms-11-02070-f003:**
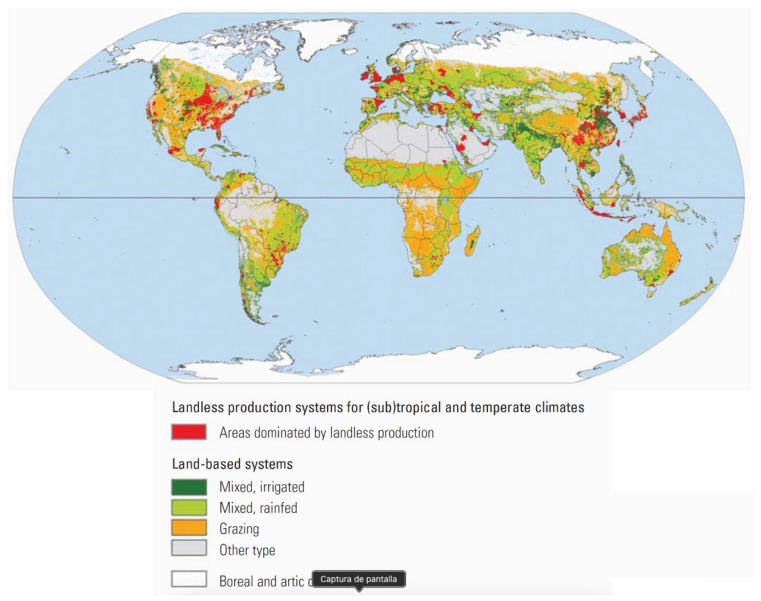
Geographical distribution of the main production systems where brucellosis-susceptible domestic livestock is raised (from [37]; reproduced with permission).

**Table 1 microorganisms-11-02070-t001:** Capacity to confront zoonotic disease challenges in 44 SSA countries ^a,b^.

	Nº (%) of Countries in Each Category	
Capacity Level [WHO JEE Score] ^c^	Surveillance Systems	Response Mechanisms
No [1]	8 (18)	17 (38)
Limited [2]	13 (30)	18 (41)
Developed [3]	19 (43)	8 (19)
Demonstrated [4]	3 (7)	1 (2)
Sustainable [5]	1 (2)	0 (0)

^a^ Adapted from [26]. ^b^ 34 countries ranked brucellosis as a priority in a list of 24 diseases. ^c^ [1] Some capacities exist but these are not organized between the animal or public health systems; no coordinated response mechanism; [2] there is a list of 5 priority zoonosis but no specific system for their surveillance or coordinated responses; [3] there are surveillance systems for 1–4 zoonoses and a multisectoral operational mechanism for responses; [4] surveillance systems exist for 5 or more zoonoses, and several experiences confirm the efficiency of the multisectoral response mechanisms; [5] there is a routine sharing of information between sectors and the multisectoral mechanism for response is regularly tested.

**Table 2 microorganisms-11-02070-t002:** PVS average scores for three areas of Veterinary Services activities relevant in One Health ^1^.

Country	Human, Physical and Financial Resources	Technical Authority and Capability	Interaction with Interested Parties
Canada	4.9	4.3	5
Japan	4.8	4.7	4.3
Chile	4.4	3.9	3.0
Uruguay	4.4	3.9	3.0
Brazil	4.0	3.2	4.5
Paraguay	3.3	2.4	3.3
India	3.2	2.7	3.0
Panama	2.8	3.1	2.8
Rep. Dominicana	2.3	2.5	2.5
Bolivia	1.8	2.6	2.5

^1^ From 5, fully developed, to 1, not developed.

**Table 3 microorganisms-11-02070-t003:** Brucellosis awareness among individuals of high-risk populations in 22 countries of Asia (6), Africa (13), South–Central America (1), North America (1), Europe (2) and the Middle East (1) ^1^.

Investigated Characteristics	Positive Answers (%)
Zoonotic nature	37.6 ^2^
Mode of transmission	35.9 ^2^
Signs of disease	
Human	41.6 ^2^
Animal	28.4 ^2^
Existence of animal vaccines	26.1 ^3^

^1^ Number of countries is in parenthesis. Adapted from [33]. Turkey (2 studies) and Palestine (1 study) were included among Asian countries. ^2^ There were no differences between the answers of individuals from better-represented areas (6 Asian countries, including China and India, and 13 African countries). ^3^ The pooled awareness level in the African population (4.6%) was notably lower than that in the Asian population (46.3%).

**Table 4 microorganisms-11-02070-t004:** Distribution of the main domestic ruminant species in different production systems ^1^.

	No. (×10^6^) in Resource-Limited Areas/Total World (%)			
Type of Animal	Grazing	Rainfed Mixed	Irrigated Mixed	Industrial
Cattle and buffaloes	342/406 (84%)	444/641 (69%)	416/450 (92%)	1/29 (3%)
Sheep and goats	405/590 (69%)	500/632 (79%)	474/546 (87%)	9/9 (100%)

^1^ Data taken from [37].

**Table 5 microorganisms-11-02070-t005:** Issues related to brucellosis characteristics.

Overlooked Facts	Consequences
Brucellosis is a collective disease; the proportion of infected herds/flocks is critical, even if individual seroprevalence is low.	Herd/flock prevalence is either not investigated or misjudged. Brucellosis presence is misrepresented and the potential of “chronically” infected herds/flocks to perpetuate the disease in an area is overlooked.
Brucellosis primarily affects animals of reproductive age.	Random selection, including young animals, reduces the detection of infected herds’ locks.
Relevance of cross-infections.	In mixed-breeding systems, focusing on one host (usually cattle): (1) does not provide the real picture, and (2) prevents control and eradication. Issue overlaps with two other mistakes: (1) *B. melitensis* does not infect cattle, and (2) *B. melitensis* infection in cattle clears spontaneously when contact with infected small ruminants is prevented.

**Table 6 microorganisms-11-02070-t006:** Diagnostic test issues ^1^.

Overlooked Facts	Consequences (Common Mistakes)
In vitro, smooth brucellae readily generate rough mutants.	For buffered agglutination ^2^ and complement fixation tests, false and inconsistent results are given. For all tests, strict quality control is necessary.
iELISA, cELISA and FPA require validation for local conditions.	Incorrect seroprevalence (use of manufacturer’s cut-offs, mostly of an unknown basis).
Sub-optimal sensitivity of cELISAs and SAT.	Seroprevalence sub-estimation.
Assays used in brucellosis-free countries for surveillance are not the best choice everywhere.	Needless infrastructure demands and costs.
Buffered acid pH agglutination tests ^1^ match iELISAs in terms of sensitivity/specificity.	It is overlooked that these agglutination tests are almost ideal under many circumstances (the misconception that these tests are negatively affected by prozones and are highly unspecific).
Misleading understanding of “Confirmatory tests”.	Incorrect seroprevalence estimations (e.g., RBT confirmed by iELISA, cELISA, FPA or complement fixation in the absence of S19 vaccination).
The milk-ring test only works in cattle (*Bos taurus*).	Incorrect prevalence estimations in small ruminants, buffaloes, and camels.
Molecular tests (PCR) require strict validation.	Unknown false-positive/negative score (identity between analytical and diagnostic parameters; “validation” in poorly defined populations (no true *Brucella*-free and no gold-standard positive controls) or in experimentally infected animals.

^1^ For detailed discussions, see [3,50]. ^2^ RBT, Card and Buffered Plate Agglutination test.

**Table 7 microorganisms-11-02070-t007:** *Brucella abortus* cattle vaccines S19 and RB51: key facts ^1^.

	Vaccine	
	S19	RB51
Used in successful programs	In 10 countries	None
Protection ^2^	*B. abortus* and *B. melitensis;* (one dose is useful throughout life).	Lower than S19 against *B. abortus*; elapses in <4 years; no evidence that revaccination bolsters protection; no evidence of protection against *B. melitensis*.
Safety		
Abortifacient	Yes (minimized by conjunctival route).	Yes.
Milk excretion	Yes (minimized by conjunctival route).	Yes.
Virulence in humans	Moderate; Standard diagnosis/treatment.	Moderate; no serological tests available; Rifampicin-resistant
Use in males	Not recommended.	Not recommended.
DIVA properties ^3^	Interferes in all serological tests; minimized in calves by the conjunctival route.	Interferes in ELISAs, FPA and lateral flow immunochromato-graphy. Animals become positive in RBT and complement fixation when exposed to virulent brucellae.
Biological control (seed lots)	WOAH (OIE) mouse model.	None.

^1^ Adapted from [48]. ^2^ Excluded the so-called “protection against abortion”, a misconception regarding true protection [48]. ^3^ DIVA, differentiation of infected and vaccinated animals.

**Table 8 microorganisms-11-02070-t008:** Brucellosis and PVS Twinning laboratories: projects completed in December 2021 ^1^.

Parent	Candidate	Year Completed
Italy	Eritrea	2011
Italy	Kazakhstan	2015
UK	Sudan	2014
France	Thailand	2013
UK	Turkey	2011
Germany	United Arab Emirates	2016
Italy	Zimbabwe	2016
UK	Afghanistan	2019

^1^ From https://www.woah.org/en/what-we-offer/improving-veterinary-services/pvs-pathway/targeted-support/sustainable-laboratory-support/laboratory-twinning/ (accessed on 3 July 2023).

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
