# Peer review of "Brucellosis and One Health: Inherited and Future Challenges"

_microorganisms, 2023, doi:10.3390/microorganisms11082070_

Round 1

Reviewer 1 Report

The manuscript titled "Brucellosis and One Health: Inherited and Future Challenges" reviews global brucellosis and impacts of disease through a One Health lens. Overall, the manuscript is well written and organized. Abstract is strong, figures are clear, and authors discuss relevant topics such as climate change and vaccination gaps and their impact on the disease. The manuscript does an excellent job discussing gaps that still exist and lays the foundation on what challenges need to be addressed with mitigating brucellosis globally. I have only minor edits to offer.

Figure 3: the text below the image appeared blurry. 

Table 8: A parenthesis is missing in the caption. 

Author Response

Reviewer 1

Comments and Suggestions for Authors

The manuscript titled "Brucellosis and One Health: Inherited and Future Challenges" reviews global brucellosis and impacts of disease through a One Health lens. Overall, the manuscript is well written and organized. Abstract is strong, figures are clear, and authors discuss relevant topics such as climate change and vaccination gaps and their impact on the disease. The manuscript does an excellent job discussing gaps that still exist and lays the foundation on what challenges need to be addressed with mitigating brucellosis globally. I have only minor edits to offer.

We are very grateful to this reviewer for the positive comments and for pointing out the defects in the original manuscript.

Figure 3: the text below the image appeared blurry. 

In the modified manuscript, we include the complete figure taken from ref. 37 to avoid any loss of resolution in any part of it. At least in our computer screen, or in a print out, this solves the problem indicated by the reviewer.

Table 8: A parenthesis is missing in the caption. 

We added the parenthesis missing in Table 8 [i.e., (accessed on 3, July, 2023)].

Again, many thanks for the work and time dedicated to our (long) manuscript.

Reviewer 2 Report

Authors have put together a well written One health review of brucellosis and existing challenges in endemic areas while also highlighting  issues that need to be addressed to help in combating the disease.

Being a  review, some sections need references to support the authors arguments. Pg 3, Pg 9.

Some statements are written in non scientific English (pg 3-end of first paragraph)and  incomplete sentence (pg 11-begining of third paragraph)

Author Response

Reviewer 2

Authors have put together a well written One health review of brucellosis and existing challenges in endemic areas while also highlighting  issues that need to be addressed to help in combating the disease.

We are very grateful to this reviewer for the positive comments and the suggestions to improve the manuscript.

Being a  review, some sections need references to support the authors arguments. Pg 3, Pg 9.

Concerning page 3, to support the statements on the interconnections among the different elements of brucellosis One Health, we refer to the Figure above, where the overlapping is clear, and also added a reference [3] where most of these aspects are discussed in the context of this zoonosis.

We also added a number of references to support the statements in page 9. Many were already in the reference list of the original manuscript, and we added 7 additional references.

Some statements are written in non scientific English (pg 3-end of first paragraph) and  incomplete sentence (pg 11-begining of third paragraph).

The text has been modified using a more academic English style (red characters in the modified manuscript).

We want to stress our gratitude for the work and time dedicated to our (long) manuscript.